# Conformation of Polyethylene Glycol inside Confined Space: Simulation and Experimental Approaches

**DOI:** 10.3390/nano11010244

**Published:** 2021-01-19

**Authors:** Tianji Ma, Nicolas Arroyo, Jean Marc Janot, Fabien Picaud, Sebastien Balme

**Affiliations:** 1Institut Européen des Membranes, UMR5635 UM ENCSM CNRS, Place Eugène Bataillon, CEDEX 5, 34095 MonFranceer, France; mtjmp2014@gmail.com (T.M.); jean-marc.janot@umontpellier.fr (J.M.J.); 2Laboratoire de Nanomédecine, Imagerie et Thérapeutique, EA4662, Centre Hospitalier Universitaire de Besançon, Université Bourgogne-Franche-Comté (UFR Sciences et Techniques), 16 Route de Gray, 25030 Besançon, France; nicolas.arroyo@edu.univ-fcomte.fr

**Keywords:** nanopore, conical, functionalization, simulations, experiments

## Abstract

The modification of the inner nanopore wall by polymers is currently used to change the specific properties of the nanosystem. Among them, the polyethylene glycol (PEG) is the most used to prevent the fouling and ensure the wettability. However, its properties depend mainly on the chain structure that is very difficult to estimate inside this confined space. Combining experimental and simulation approaches, we provide an insight to the consequence of the PEG presence inside the nanopore on the nanopore properties. We show, in particular, that the cation type in the electrolyte, together with the type of electrolyte (water or urea), is at the origin of the ion transport modification in the nanopore.

## 1. Introduction

Polyethylene glycol (PEG) is a hydrophilic polymer that is commonly used for surface treatment [1]. It allows to ensure the surface wettability and the protection against corrosion [2,3]. In sensor area, PEG is often used as a spacer to graft proteins on the solid-state surface [4,5]. The extensive use of PEG comes from its low affinity to bind proteins [6]. More generally, PEG is one useful way to provide antifouling properties of a membrane [7]. It prevents the deposition of biomacromolecule as well as biofilm [8,9]. Whatever the surface, the properties of PEG coating depend on its conformation and on the molecular weight of the polymer chain [10]. Typically, a PEG size between 2 kDa and 10 kDa is optimal to prevent the protein adsorption [11]. However, the polymer conformation and thus the grafting density are likely the most important parameters to enhance antifouling properties [12,13]. The brush conformation induced by a high density of grafting is the most efficient configuration to prevent the protein adsorption. The polymer conformation on a surface is given from several scaling laws. The Flory’s law allows calculating the polymer radius under good solvent condition since the surface density is weak.
(1)$$R_{F}=N^{3/5}a$$
where *N* is the number of polymer units and *a* the size of one monomer. Under ideal or bad solvent conditions, the polymer radii (*R*_0_ and *R_B_*) are obtained by
(2)$$R_{0}=N^{1/2}a$$
(3)$$R_{B}=N^{1/3}a$$

At high grafting density, the PEG adopts a brush conformation. In that case, it exists a relation between the thickness *H* and the grafting density *Γ*.
(4)$$H={\left(\frac{\Gamma }{3}\right)}^{1/3}b^{1/2}aN$$
where *b* is the Kuhn length.

The antifouling and surface protection properties of PEG make it a good candidate to functionalize solid-state and polymer nanopore for sensing applications. Indeed, the main limitations of a silicon nitride (SiN) nanopore are its short lifetime and its poor stability. The chemical grafting of PEG considerably improves SiN nanopore lifetime [14]. It also allows detecting protein aggregates without fouling [15]. PEG coating was also considered to prevent the fouling of track-etched nanopore during the detection of β-lactoglobulin [16,17] and of protein Tau [18]. The PEG is also used as spacers to provide photo-switch properties [19] as well as to build protein layers’ architecture [20]. More generally, polyelectrolytes were widely employed to provide stimuli-responsive properties with pH or electrolyte [21,22]. The Poly(N-isopropyl acrylamide) allows the design of thermal responsive nanopore [23,24]. In the cell, the FG domain of the nuclear pore ensures the exceptional selectivity because only the macromolecule bound to a cargo can cross the pore [25,26].

Generally speaking, the confinement of macromolecules inside nanoporous material such as Al_2_O_3_ or mobile composition matter (MCM) was extensively investigated in the lack of a solvent [27]. Typically, it was demonstrated that increasing the constraints by decreasing the pore size enhanced the intermolecular interaction and strongly modified their properties such as the glass transition temperature [28,29]. In addition, both the interfacial energy and the wettability play a significant role on the polymer dynamic inside the nanopore [30] as well as the molecular weight of the polymers [31]. With respect to the literature, it appears clear that the polymer behavior under constraint involves complex phenomena where numerous parameters influence their structures. According to that and considering the addition of an electrolyte solution, the question about the impact of confinement on the polymer conformation should be well understood since it is the keystone of the ionic transport properties. This will help the experimentalist to optimize the design of polymer functionalized nanopore.

In recent literature, the PEG was grafted inside SiN nanopore with a diameter larger than PEG thickness under good solvent conditions. In these cases, it was reported a mushroom conformation [15]. In addition, the chaotropic or kosmotropic properties of the electrolytes modify the nanopore diameter [14]. Typically, in LiCl solution, the PEG adopts a conformation like ideal solvent while in KCl, the conformation is closer to poor solvent [32]. The following question is what happens when the nanopore diameter is smaller than the PEG radius under good solvent conditions. In other words, how the effect of confinement influences the PEG conformation. This question was previously investigated from a theoretical point of view. It was reported that polymers tend to escape from the nanopore in good solvent and bend into solvent in poor solvent conditions [33]. Considering the nanopore functionalization by PEG, the latter conformation is not the unique unsolved question. Indeed, the grafting density was never reported due to difficulties to characterize the ultimate thin inner volume at single nanopore scale. However, this information is essential to evaluate the PEG density onto the nanopore surface wall. For instance, it is required to compare experimental measurements with theoretical models as well as to model the system for molecular dynamic simulation. In addition, considering the specific case of conical nanopore, the ionic current rectification could be strongly affected by the polymer conformation, location and density [34,35,36]. The PEG is uncharged, and we could expect that if the grafting involved all carboxylate groups, it would avoid the current rectification.

This work aims to investigate the effect of confinement on the PEG conformation. To do so, we designed conical nanopore with tip diameter smaller than the PEG size under good solvent condition. First, we evaluated the grafting density of PEG since this question is essential for further investigations. Then, the PEG size inside the nanopore was measured under chaotropic or kosmotropic electrolyte conditions as well as with the presence of urea. The urea is one of the most denaturant agent uses to denaturate a protein. However, it is not always taken into account as an agent that modifies the electrolyte transport and thus the ionic current. The experimental results are completed by molecular dynamic simulation investigation to provide a complete description of the confinement effects of PEG. Besides that, the ionic current rectification is also discussed since it allows making an idea about the charge regulation inside the nanopore.

## 2. Materials and Methods

### 2.1. Materials

We began with 13 μm thick PET films, with biaxial orientations purchased from Goodfellow (San Dieguito, CA, USA) (ref. ES301061). Amine-PEG (PG1-AM-5K) was purchased from Nanocs (New York, NY, USA). 1-Ethyl-3-(-3-(dimethylamino)propyl) carbodiimide hydrochloride (EDC, E7750), MES hydrate (M8250), sodium chloride (NaCl, 71380), HEPES (H3375), urea (U5378) and trimethylamine N-oxide dehydrate (TMAO, 92277) were purchased from Sigma-Aldrich (St. Louis, MO, USA). Potassium chloride (KCl, POCL-00A-1 k0) was obtained from Labken. Chloride acid (20248.290) and sodium hydroxide (28245.298) were purchased from VWR Chemicals (Radnor, PA, USA). Water used in these experiments was purified by Q-grad^®^-1 Milli-Q system (Millipore, Burlington, MA, USA). 

### 2.2. Single Track-Etched Nanopore Fabrication and Functionalization

Single tracks were produced by Xe irradiation (8.98 MeV/u) in PET film at GANIL, SME line (Caen, France). A hole (diameter 1 mm) with a shutter was placed on the ion beam path. A detector placed behind the sample controlled the track number. PET films with single tracks are stocked in aluminum paper and sealed in plastic bags. They are taken out the day before etching. Firstly, a piece of PET film (2 cm × 2 cm) with a single track was activated by UV irradiation (Fisher bioblock; VL215.MC, l = 312 nm) during 9 h per side. For the conical nanopore, the PET film was then mounted in a Teflon cell with two chambers with an etchant solution (9 M NaOH) and a stopping solution (1 M KCl and 1 M acetic acid). Pt electrodes were used to apply electrical bias of 1 V across the film to control the pore opening by measuring the current as a function of time using an amplifier (HEKA EPC10). Then, the etching process was stopped by replacing the etchant solution with stopping solution and finally, the membrane was immersed in 18.2 MΩ cm pure water for 24 h to get cleaned. The diameter of single nanopore is calculated from the dependence of the conductance G measured in the linear zone of the I–V curve (−60 mV to 60 mV) in 1 M NaCl solution:(5)$$G=\frac{\kappa dD\pi }{4L}$$
where, *κ* is the ionic conductivity of the solution assuming a bulk-like transport, *L* is the nanopore length (13 μm), *d* and *D* the tip and base diameter respectively. *D* is calculated from the total etching time *t* using the relationship *D* = 2.5*t* (the factor 2.5 is determined in our experimental set-up using multipore track-etched membranes) [37].

The cylindrical nanopores were obtained by immersion of a single track in NaOH (3 M) at 50 °C during a certain time (between 4 to 15 min) to reach the expected diameter. Then the nanopore was rinsed several times and left in deionized water one night before using.

### 2.3. PEG Functionalization

The PEG grafting inside nanopore immobilization was as follows. A mixture of 0.1 M MES hydrate, 0.1 M KCl, 50 mM EDC and a point of spatula of Amine-PEG-5K at pH 4.7 is added in chambers of both sides of the nanopore. The reaction was left overnight. Then, the nanopore was washed several times.

### 2.4. Current-Voltage Measurements

The current-voltage measurements were performed using a patch-clamp amplifier (EPC10 HEKA electronics, Germany) with Ag/AgCl electrodes. Current traces were recorded as a function of time from 1 V to −1 V by steps of 100 mV during 2 s each for rectification factor calculation and from 100 mV to −100 mV by steps of 10 mV during 2 s each for conductance calculation. These electrical measurements for characterizing PEG immobilized conical nanopore were performed in solutions with salt electrolytes (KCl, or LiCl with varying concentrations) and 5 mM HEPES at pH 7.4. For characterizing urea effects, the latter compound was added with respective concentrations in previous solutions always with 5 mM HEPES at pH 7.4.

### 2.5. Molecular Dynamic Simulation

Classical all-atom MD simulations were performed using the NAMD.2.12 package [38]. The different systems were solvated in a water box large enough to prevent the interaction between the central part (the conical pore) and its neighboring periodic cells. KCl ions (at a concentration of 1 M) were added to the water (simulated using the TIP3P model) to reproduce the experimental environment. CHARMM36 [39,40] force-field optimization parameters were used in all simulations. During the simulations, the system temperature and pressure were kept constant at 300 K (Langevin dynamics) and 1 atm (Langevin piston), respectively. The long-range electrostatic forces were evaluated using the classical particle mesh Ewald (PME) method with a grid spacing of 1.2 Å, and a fourth-order spline interpolation. The integration time step was equal to 1 fs. Each simulation employed periodic boundary conditions in the three directions of space.

### 2.6. Construction of the Functionalized Conical Nanopore

To build the conical solid-state nanopore (*R_tip_* = 3 nm, *R_base_* = 3.5 nm, length = 14.8 nm), several carbon nanotube sections of different radii were associated and centered along the nanopore axis. To model the chemical structure of PET nanopore used in the experiments, one third of the carbon atoms constituting the nanopore were randomly configured as oxygen atoms. Two graphene sheets are used to simulate the membrane. Every element of those nanostructures is stationary, to simulate the fact that the nanopore is in fact a hole in a membrane, hence with rigid walls. Partial charges, positive for carbon atoms and negative for oxygen atoms, were added while global neutrality of the nanopore was conserved.

We then functionalized the nanopore according to the experimental functionalization by grafting 30 hydroxyls functions to randomly-picked carbon atoms. Then, three PEG 5K were added (for a grafted density of 0.01 PEG nm^−2^), as in the experiments. To place the PEG, a position of linkage was chosen with a random generator number. Nevertheless, to orient each molecule, we took into account the position of each one in order to avoid the superposition of the different structures. 

Finally, the functionalized nanopore was placed between 2 reservoirs of dimensions equal to 13.3 × 13.3 × 6.8 nm^3^ and solvated with either K^+^ or Li^+^ and Cl^−^ ions concentration equal to 1 M. The total reservoir size was chosen to be around twice the size of the conical-tube part of the nanopore. The complete system dimensions are 13.3 × 13.3 × 28.5 nm^3^, for a total of around 330.000 atoms, detailed as such: 24.000 for the nanopore, 2.400 for PEGs, 3.700 for ions and 290.000 for water. We also modeled another solvent, similar to the previously described ones, but with urea molecules at high concentration (4 M). Those four systems will allow us to compare the effects on PEG’s conformation and radial distribution of the different components, considering different ions in the solvent and the presence of urea molecules. 

Once each complete functionalized and solvated system was assembled, it was then optimized following three successive procedures. First, we minimized the energy of the total system at 0 K. Then, the system was progressively heated until reaching a temperature equal to 300 K. Finally, the system was left to evolve at the NPT ensemble, and physical observables were calculated using time averages (See Figure 1). During all simulations, every atom constituting the conical nanopore was kept fixed. Systems without urea were left to evolve for 25 ns, and systems with urea for 40 ns.

## 3. Results

### 3.1. Determination of the Surface Coverage

The conical nanopores were obtained by track-etched technique following the electrostopping procedure [36]. The nanopore dimension as well as the surface charge can be directly obtained from the conductance (G) as a function of the salt concentration.
(6)$$G=\kappa_{B}\frac{\pi rR}{L}+\mu_{+}\sigma \frac{\left(r+R\right)/2}{L}$$
where *G* is the ionic conductance, *R* the radius, and *L* the length of the pore. *σ* is the nanopore surface charge, and $\kappa_{B}=e^{2}\left(\mu_{+}+\mu_{-}\right)c_{B}$ is the bulk conductivity, *e* is the elementary charge, and *μ* the ion mobility. At high electrolyte concentration, the second term of the equation is negligible. In that case, the conductance electrolyte concentration dependence is a direct function of the nanopore radius. Conversely, at low salt concentration, the surface charge leads the nanopore conductance, and thus, the conductance becomes constant. The surface charge is ensured by the carboxylic acid moieties due to the chemical etching that reacts with PEG-NH_2_. After PEG grafting, we observe a decrease of the nanopore conductance as well as of the plateau value at low salt concentration (Figure 2a,b). A rough estimation of the plateau value ratio before and after PEG grafting gives a decrease of about 60% of the surface charge. However, this estimation could be far from the reality since the nanopore is conical and the steric effect is not the same at the narrowest aperture of the pore. This could induce the heterogeneity of the PEG functionalization. In order to investigate that, we studied the cover rate of PEG-NH_2_ for cylindrical nanopore with radii ranging from 8 to 100 nm (Appendix A). We extracted from the conductance vs. concentration curves, the nanopore radius and the surface charge before and after PEG grafting using the analytical model previously proposed by Balme et al. (see Equations (13) and (14) of Reference [37]). From the ratio of surface charge after and before PEG grafting and considering that one PEG-NH2 molecule is grafted to one COO-, we obtained a surface density of PEG. We observe in Figure 2c the surface density of PEG on the nanopore radius. Typically, for a radius below 20 nm, the PEG density increases with the nanopore diameter and decreases for larger nanopore to reach a constant value of about 0.12 PEG nm^−2^. We notice the presence of a maximum for the nanopore with a radius of about 20 nm. The lower cover rate for small nanopore can be explained by a steric effect. The passage to a maximum could be due to a difference of nanopore surface wall structure in the irradiation halo. This was previously reported by Dejardin et al. in the case of polycarbonate track-etched membrane [38].

### 3.2. PEG Conformation Impact of Confinement

Now, we investigate the effect of the electrolyte solutions on the PEG conformation inside a small nanopore (*r_tip_* = 5 nm and *r_base_* = 100 nm). After chemical grafting of PEG 5 kDa, the IV curves were recorded at various LiCl and KCl concentrations from 10^−4^ M to 2 M (Figure 3a). The conductances measured from the linear zone of IV curve (from −100 mV to 100 mV) are reported as a function of the salt concentration on Figure 3b. We can notice that there is not significant difference between LiCl and KCl at high salt concentration. The calculation reveals that the PEG thicknesses are similar ~4 nm corresponding to a PEG conformation under ideal solvent condition. This result is totally counterintuitive since it was previously demonstrated for PEG grafted inside SiN nanopore, that under LiCl, the PEG chain adopts an ideal solvent conformation, while under KCl, it behaves as a poor solvent one [32]. The main difference here is the size and the chemical structure of the nanopore. The effects of chaotrope and komostrope cations are well known under bulk solution conditions. However, on surface, the interaction PEG/material should be taken into account. In order to know if the observed results are due to the strong confinement or the chemical properties of nanopore, we performed similar experiments with larger nanopore (*d_tip_* = 9 nm and *D_base_* = 123 nm). Under this condition, the PEG conformation is less physically constrained. The measurement of the PEG thickness under LiCl is about 8 nm and KCl 3 nm. We now compare our experimental results with the theoretical size of PEG under good, ideal or poor solvent and brush conformation (Equation (4)). For the latter, we plot in Figure 3c the PEG thickness as a function of the *surface density* (expressed in % PEG coverage), calculated using a surface charge extracted from the cylindrical nanopore 0.335 ± 0.1 e^−^ nm^−2^ and the ratio of PEG grafting. For the larger nanopore, we estimate from Figure 2c a surface density of PEG equal to about 0.11 PEG nm^−2^ corresponding to ~30% of surface coverage. In this case, the PEG size obtained under LiCl is between the good solvent regime (Equation (1)) and a brush conformation. Conversely, under KCl the PEG chains are collapsed. For the smaller nanopore, the extrapolation of the surface density of PEG from Figure 2c gives about 0.01 PEG nm^−2^ corresponding to ~3% of surface coverage. Here, the low density of PEG can explain the weak difference of PEG thickness obtained under LiCl and KCl.

To further investigate the PEG conformation inside the nanopore, we performed molecular dynamic simulations. The nanopore dimensions (*R_tip_* = 3 nm, *R_base_* = 3.5 nm, length = 14.8 nm) are chosen to be as close to the tip side of the experimental small pore. The number of PEG grafted inside the nanopore was determined to be equal to 3 (related to the density of 0.01 PEG nm^−2^) according to the extrapolation from the experimental results (Figure 2c). In Figure 4 are combined the distributions of the atoms to see the whole organization inside the pore. In this graph, reduced variables for the distance have been chosen, i.e., 0 represents the center of the nanopore and 1 is for the surface wall. Going from the outside, we find cations Li^+^ at 0.95R (3.08 nm) and K^+^ at 0.93R (3.00 nm). Regardless the cation in the simulation, we then observe the water peaking at 0.91R (2.97 nm) and the anion (Cl^−^) at 0.90R (2.93 nm). Except for the ions themselves, the cation ion in the simulation makes no difference in the organization of the first peak observed for each component of the system. The distribution of PEGs shows nearly no difference in behavior since, regardless the cation, the PEG peak is around 0.875R (2.86 nm), and few atoms are detected before 0.3R. We, although, observe that slightly more PEG is found close to the nanopore center with the LiCl solvent. This seems to indicate that PEG, while being only attached by one atom to the pore, still globally stays in its vicinity and is not expanding to the full volume available. We also note that in the initial frame of the simulation, distribution peaks around 0.05R. This shows that PEG converged to its stable position close to the pore wall despite being in a starting configuration which was quite far from the equilibrium position.

### 3.3. Impact PEG Grafting on the Ionic Current Rectification

We have previously shown that the PEG grafting does not avoid all carboxylate groups. This surface charge is at the origin, for low salt concentration, of a current rectification. The values of current measured at +1 V and −1 V and the ratio named rectification factor (Rf) are reported in Figure 5. For KCl and LiCl, the Rf values follow the same trend. For a salt concentration below 10^−2^ M, the rectification factors are found below 1. This means that the nanopore is selective to cations. This is in accordance with the negative surface charge due to the carboxylate moieties. When the salt concentration reaches about 5 × 10^−2^ M, Rf suddenly increases until a value close to 1. Thus, the nanopore loses its selectivity properties. It is interesting to notice that after the PEG grafting, the nanopore diameter is reduced from 5 nm to 1 nm. The loss of selectivity occurs when the Debye length is close to the nanopore diameter. 

### 3.4. Effect of Urea on PEG Conformation and Ionic Current Rectification

In a previous work, Roman et al. reported the impact of urea on KCl and LiCl conductance through PEGylated nanopore [27]. They showed, at high salt concentration, the effect on the increase of solution viscosity for both LiCl and KCl. We did similar experiments using the small conical nanopore. The experiments were conducted for two different electrolytes at two concentrations: 1 M and 0.01 M. At 1 M of LiCl and KCl, the conductance linearly decreases with the urea concentration. This could be assigned to an increase of the solution viscosity. In addition, the rectification factor is constant. For lower concentration (0.01 M), the conductance also decreases with the urea concentration. Below urea 4 M, we can observe a large current rectification. Upper this concentration, the nanopore seems to be closed (Figure 6). The nanopore enclosure is sharper for KCl than LiCl suggesting a cooperative effect of two chaotropic entities.

The experimental results of the nanopore conductance suggested that the urea does not modify strongly the conformation of the PEG chain. However, it could influence the organization of the PEG molecules inside the pore by taking the place of water molecules inside the nanopore. This is plausible since the polymer mushroom structure is compact due to low surface density of PEG. In order to confirm that, we analyze the atom location thanks to molecular dynamic simulation.

First, while the initial distribution of urea is rather homogenous in the two-third radius pore (to avoid collisions with the surface during the construction phase as shown in Figure 7), it evolves into a sharp peak at 0.9R. Urea seems to heavily interact with the PET surface, as shown in Figure 8. Urea behavior shows no difference whether KCl or LiCl is present in the system.

In presence of urea, we observe comparable distribution of atoms except a shift of the PEG peak from 0.875R to 0.795R. This can be explained by the presence of urea close to the pore surface (0.9R), pushing PEGs away from it. We also observe here an influence of the cation on the PEG structure. Indeed, the polymer chains take more space in the center of the pore under LiCl than KCl; the minimum radius at which PEG atoms are found are around 0.35R for KCl and 0.1R for LiCl (see Appendix A for clarity). 

Interestingly, we did not observe any significant difference in the shape of water distribution in regard to the presence of urea or the type of cation (see Appendix A). Indeed, we observe a high peak of water at 0.91R for all the systems near the pore surface and a first layer of positive ions. Then, a small drop in water density at 0.87R is appearing just after the first peak (Figure 7). This second peak is less observable with urea. Then, the water distribution is homogenous in the center of the pore. It seems that the presence of PEG is at the origin of this water distribution. The two peaks are present with or without urea and the only modification in the water distribution with urea comes from the intensity of these peaks. With urea, several water molecules have been ejected from the pore surface, leaving the place to urea molecules (as shown in Figure 8 which depicts the time evolution of the pair interaction with the pore wall of the water and urea molecules).

To discuss the effect of urea on PEG’s conformation, we plot the RMSD of the complete PEG structures over time for each system (Figure 9). We observe that urea (4 M) does have a noticeable effect on PEG’s RMSD. We compute the average RMSD of the last 5 ns (15 ns) for systems without (with) urea. As such, we obtain values without urea of 21.7 ± 0.7 Å (KCl) and 21.0 ± 0.5 Å (LiCl), and with urea 18.6 ± 0.3 Å (KCl) and 18.3 ± 0.7 Å (LiCl). This is consistent with the distributions already discussed. PEGs start the simulation closer to the center of the tube (Figure 10). They end up being more distant from the surface because of the layer of urea close to the pore surface (Figure 4, Figure 7 and Figure 10), thus, the smaller RMSD obtained in presence of urea. Nevertheless, the PEG morphology (as shown in Appendix A) has been slightly modified in the nanopore with the flattening of the secondary peak density at 0.7R with urea. This could interpret the conductance change observed in the experiment when the urea is incorporated in the solvent.

## 4. Conclusions

The chemical functionalization of a conical nanopore by PEG has been studied using both experimental and simulation works. Several electrolyte conditions have been envisaged, giving a complete overview of the specific organization of the PEG inside the nanopore and its consequence on the properties of the nanopore. Our experimental measurements revealed that the PEG structure under strong confinement cannot be predicted by Flory’s law. In addition, the ion does not affect the chain structure conversely when the nanopore is larger. However, the PEG affects the ion transport inside the nanopore, depending on the cation type and the presence of urea. This has been interpreted by molecular dynamic simulations as coming from the specific extension of the PEG molecules inside the pore which is more pronounced with LiCl compared to KCl. Regarding the growth interest of PEG uses in the nanopore sensing experiment, we believe that our work will provide a better understanding of the polymer conformation and its impact on the ionic transport.

## Figures and Tables

**Figure 1 nanomaterials-11-00244-f001:**
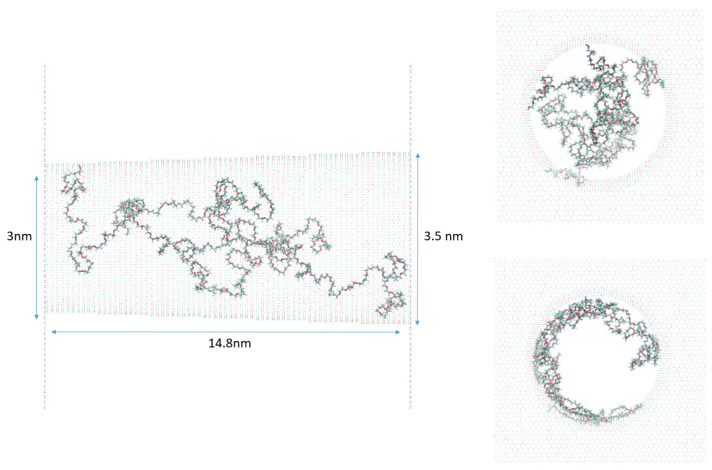
On the left and upper right, respectively, a profile and a front view of the starting point of simulations. On the lower right is shown the front view of the KCl system without urea after 20 ns of simulation. Only PEG (polyethylene glycol) and the pore are represented here.

**Figure 2 nanomaterials-11-00244-f002:**
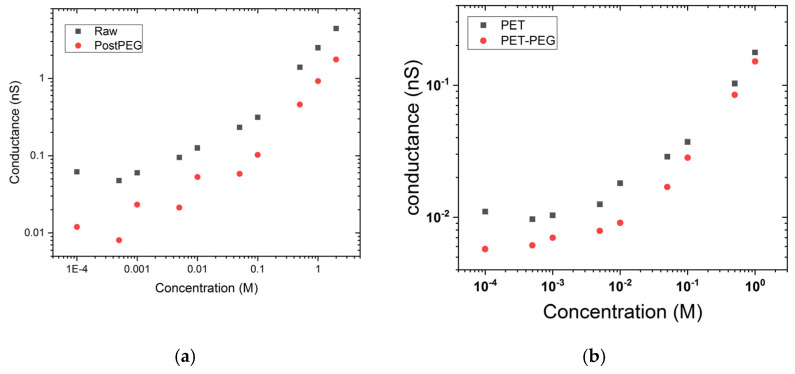
Conductance as a function of KCl concentration for (**a**) conical nanopore (*d_tip_* = 10 nm, *D_base_* = 200 nm) and (**b**) cylindrical nanopore (*d* = 19 nm) before (black square) and after (red, circle) PEG functionalization. (**c**) Evolution of cover rate of PEG calculated from the decrease of conductance at low salt concentration for cylindrical nanopores.

**Figure 3 nanomaterials-11-00244-f003:**
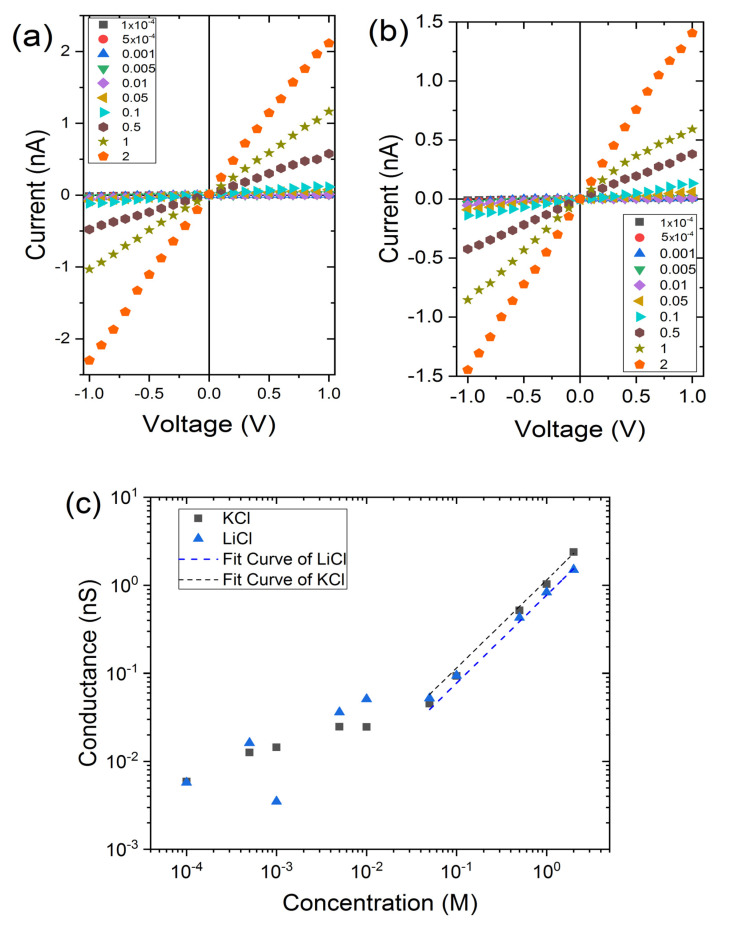
IV curves obtained for the small conical nanopore (*d_tip_* = 10 nm, *D_base_* = 200 nm) under LiCl (**a**) and KCl (**b**) solution; (**c**) Nanopore conductance as a function of KCl and LiCl concentration, the dash line are the linear fits at high salt concentration to measure the nanopore diameter; (**d**) Map depicting the measure PEG layer thickness for small nanopore (square) and large nanopore (circle); the dash line are the expected value obtained from Flory’s law and Equation (4) assuming a maximum surface density of 0.33 PEG nm^−^².

**Figure 4 nanomaterials-11-00244-f004:**
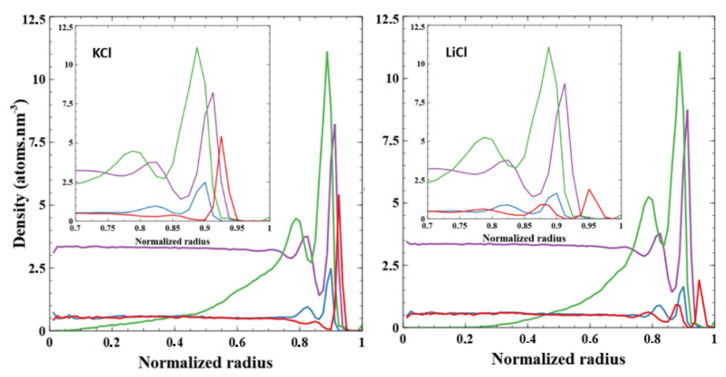
Radial distributions of every component inside the tube after 25 ns of simulation averaged from the last 5 ns. Cations (either K^+^ or Li^+^) are shown in red and Cl^−^ in blue. PEGs are shown in green. Water is represented in purple, its density in the graph is divided by 10 to improve clarity. KCl solvent is shown on the left graph while LiCl is shown on the right.

**Figure 5 nanomaterials-11-00244-f005:**
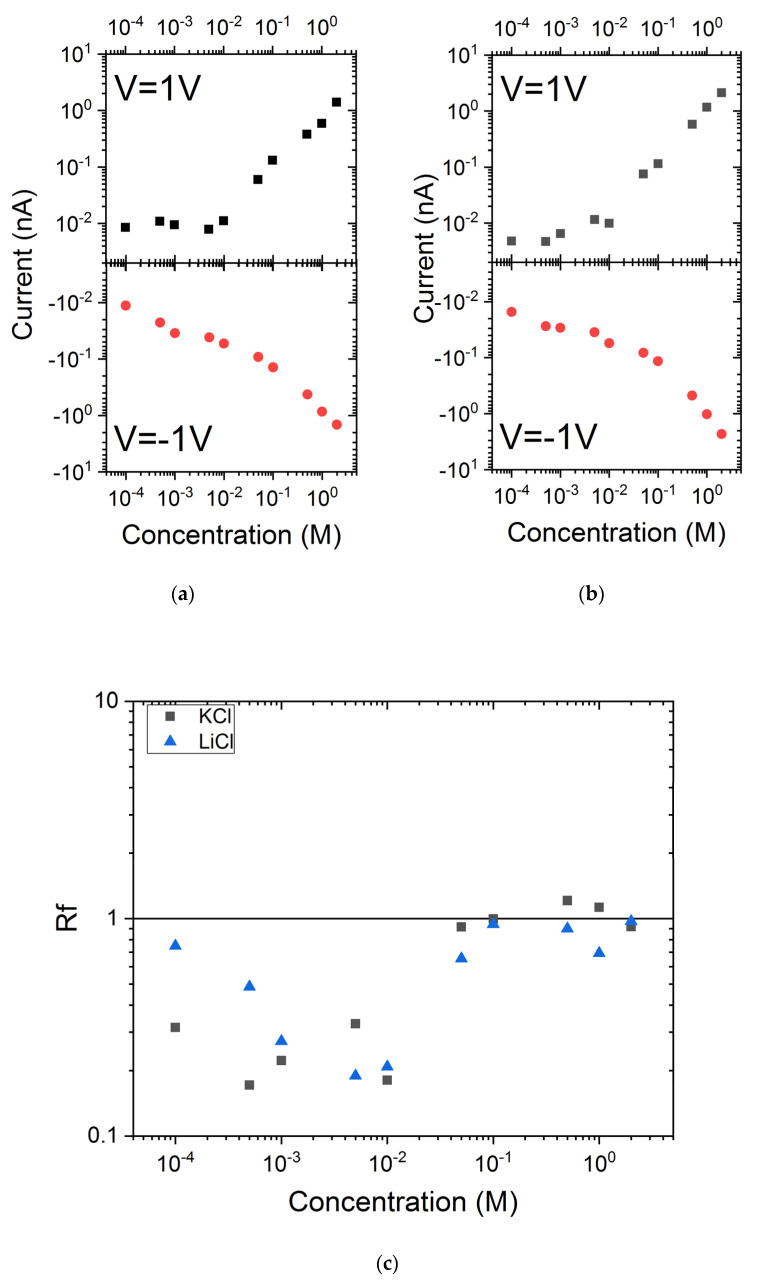
Current under 1 V and −1 V recorded under LiCl (**a**) and KCl (**b**) solution; (**c**) Evolution of ionic current rectification as a function of electrolyte concentration.

**Figure 6 nanomaterials-11-00244-f006:**
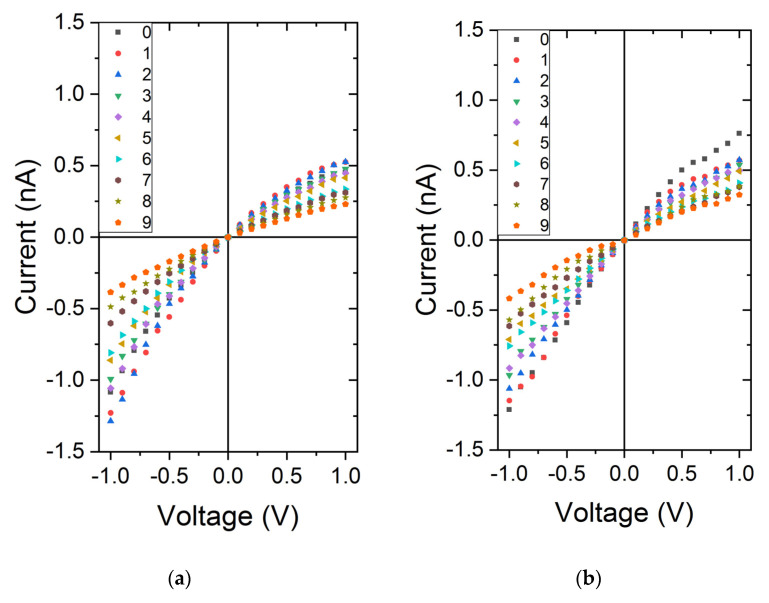
IV curves obtained for the small conical nanopore (d*_tip_* = 10 nm, D*_base_* = 200 nm) under KCl 1 M (**a**) and 0.01 M (**b**) and LiCl 1 M (**d**) and 0.01 M (**e**) solution (different colors have been chosen as a function of the urea concentration as shown in the inset caption); Nanopore conductance as a function of urea concentration KCl and LiCl 1 M (**c**) and 0.01 M (**f**); (**g**) Evolution of ionic current rectification as a function of urea concentration.

**Figure 7 nanomaterials-11-00244-f007:**
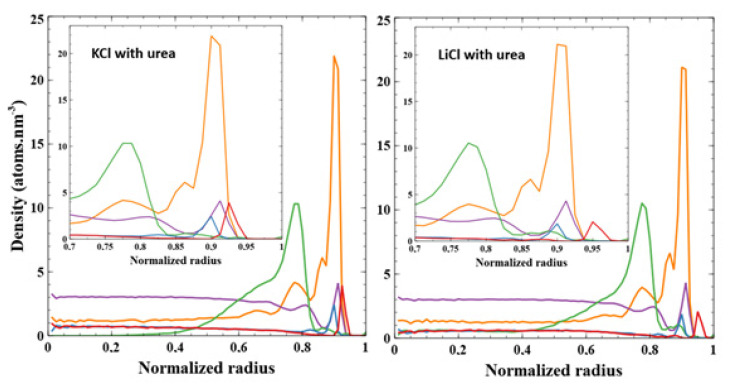
Radial distributions of every component inside the tube after 40 ns of simulation averaged from the last 5 ns. Cations (either K^+^ or Li^+^) are shown in red and Cl^−^ in blue. PEGs are shown in green. Water is represented in purple, its density in the graph is divided by 10 to improve clarity; urea is shown in orange. KCl solvent with urea is on the left graph while LiCl with urea is on the right.

**Figure 8 nanomaterials-11-00244-f008:**
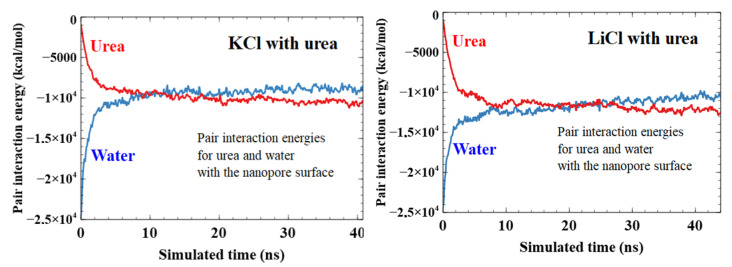
Pair interaction energy in kcal/mol between both water and urea and the nanopore surface (not including graphene sheets) over the duration of the simulation. We see how urea/surface energy decreases (i.e., attraction increases) while water/surface energy increases.

**Figure 9 nanomaterials-11-00244-f009:**
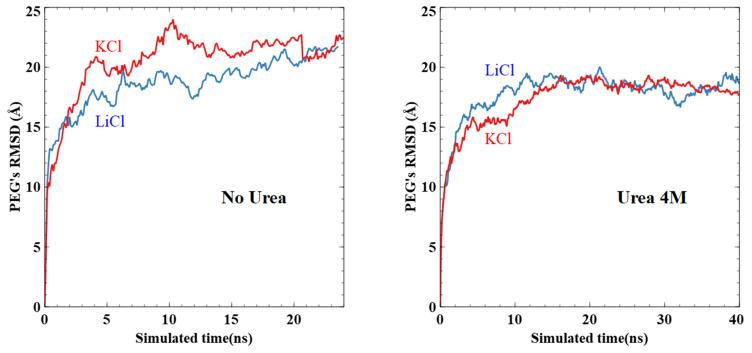
RMSD of PEG structures from the beginning of simulations without (**left**) and with (**right**) urea. Systems with KCl are shown in red while LiCl is in blue.

**Figure 10 nanomaterials-11-00244-f010:**
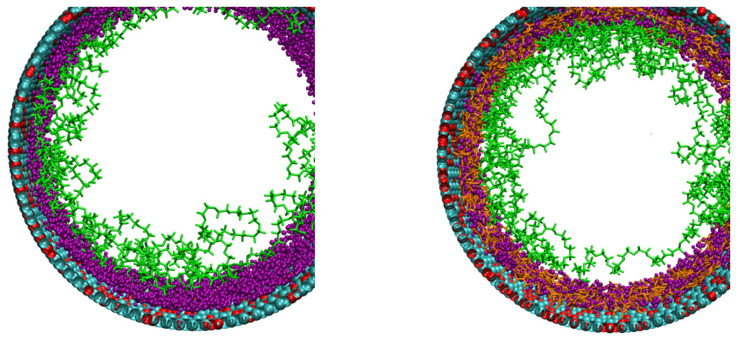
Front view visualization of the KCl systems, without (**left**) and with (**right**) urea after, respectively, 25 and 40 ns of simulated time. PEG, water and urea near the pore surface are shown in green, purple and orange, respectively. Nanopore is shown in teal and red.

## Data Availability

The data presented in this study are available on request from the corresponding author.

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
