# Peer review of "Conformation of Polyethylene Glycol inside Confined Space: Simulation and Experimental Approaches"

_nanomaterials, 2021, doi:10.3390/nano11010244_

Round 1

Reviewer 1 Report

The authors wanted to investigate the effect of confinement of PEG conformation. They evaluated the grafting density of PEG in the conical nanopore and then they measured the PEG size inside under chaotropic or kosmotropic electrolyte condition as well as with the presence of urea. The conclusions are incomplete but the authors remarqued that the ion does not affect the chain structure conversely when the nanopore is larger and the PEG affects the ion transport inside the nanopore, depending on the cation type and the presence of urea.  

Issues: 

  1. Authors should more clearly explain the effect of urea on the conformation of PEG chain;
  2.  Line 95-96: Potassium chloride 95 (KCl, POCL-00A-1 k0) was obtained (acquired) from Labken.

    Line 105-106: was activated by UV irradiation (Fisher bioblock; VL215.MC, l= 312 nm) during 9 one night ?????

    Line 205: eq 5 become 6

    Line 214: what figure (Figure a)???

    Line 227: of about 0.12 PEG. nm-2. Rewrite correctly... author is not consistent with the measure units: nm-2 or nm-2

Author Response

Referee 1

The authors wanted to investigate the effect of confinement of PEG conformation. They evaluated the grafting density of PEG in the conical nanopore and then they measured the PEG size inside under chaotropic or kosmotropic electrolyte condition as well as with the presence of urea. The conclusions are incomplete but the authors remarqued that the ion does not affect the chain structure conversely when the nanopore is larger and the PEG affects the ion transport inside the nanopore, depending on the cation type and the presence of urea.  

We thank the referee for his/her positives comments

Issues: 

  1. Authors should more clearly explain the effect of urea on the conformation of PEG chain;

Our answer: we have amended the new version of the manuscript with a description of the PEG absolute position and deformation. 2 new figures (9 and 10) were also added to justify this discussion (see last paragraph before conclusion line 439-450))

  1. Line 95-96: Potassium chloride 95 (KCl, POCL-00A-1 k0) was obtained (acquired) from Labken.

Line 105-106: was activated by UV irradiation (Fisher bioblock; VL215.MC, l= 312 nm) during 9 one night ?????

Line 205: eq 5 become 6

Line 214: what figure (Figure a)???

Line 227: of about 0.12 PEG. nm-2. Rewrite correctly... author is not consistent with the measure units: nm-2 or nm-2

Our answer: We thank the referee to point several mistakes along our manuscript. All of them were corrected.

Reviewer 2 Report

Recommendation: This paper should be considered for publication in Nanomaterials after revision.

The manuscript by Ma et al entitled “Conformation of Poly(ethylene glycol) inside confined space: simulation and experimental approaches” is devoted to study behavior of PEG inside pores of varying diameter, relationship between current characteristic and conformation of this polymer attached to the surface of the pore walls. Generally manuscript is quite interesting and gives nice contribution to better understanding of charge transport in pores. However, there are some concerns that should be addressed prior publication of this paper in Nanomaterials.

  • Authors claim that they studied conformation of PEG, but as far as I see the problem of varying conformation was addressed using molecular dynamics simulations data. Why not to use more direct methods such as Infra red or Raman spectroscopies to touch this problem? In fact in papers by Kremer and Paluch groups it was shown that application of these techniques allows to address this issue very well, please see following papers Kipnusu et al Soft Matter,9, 4681-4686 (2013), Kipnusu et al , Zeitschrift für Physikalische Chemie 226, (2018), Tarnacka et al J. Phys Chem. C, 122, 28033-28044, (2018), Minecka et al Chem. Chem. Phys., 20, 30200-30208 (2018). I would extend introduction section to put the context of variation in molecular conformation in low and high molecular weight systems confined in pores in wider perspective as quite important problem. That for sure would strengthen the outcome of the manuscript.
  • The other quite obvious issue that may explain data reported by Authors is variation in wettability of urea, water or PEG close to the pore walls. Did Authors consider measuring contact angles to correlate their results with wettability. Herein it should be noted that Floudas group (Alexandris et al Macromolecules 2016, 49, 7400−) and further Talik et al (JPC C 123, 5549−5556, 2019) have shown that interfacial energy, wettability are two important factors controlling dynamics and most likely conductivity (it is directly connected to the mobility of the ions and structural/segmental dynamics).
  • Authors did not reference to quite large and significant articles devoted to studying charge transport in pores (see papers by Kremer or Singh groups for example Iacob et al Soft Matter 2012, 8, 289− Tripathi et al J. Mater. Chem. A 2015, 3, 23809−23820.). I think additional discussion in view of these papers would be useful for the Authors.

Less important /minor points

  • Authors in many places use SiN acronym for pores. Please defined what SiN stands for.
  • Authors should improve quality of the Figures they are not readable.
  • Authors should provide more explicit explanation why did they use urea not other agents?
  • Under equation 1 authors did not define what a stands for. I think it should be size of one monomer. Please correct it.

After addressing following issues I do recommend publication of the paper in Nanomaterials.

Author Response

Referee 2

The manuscript by Ma et al. entitled “Conformation of Poly(ethylene glycol) inside confined space: simulation and experimental approaches” is devoted to study behavior of PEG inside pores of varying diameter, relationship between current characteristic and conformation of this polymer attached to the surface of the pore walls. Generally manuscript is quite interesting and gives nice contribution to better understanding of charge transport in pores. However, there are some concerns that should be addressed prior publication of this paper in Nanomaterials.

  • Authors claim that they studied conformation of PEG, but as far as I see the problem of varying conformation was addressed using molecular dynamics simulations data. Why not to use more direct methods such as Infra red or Raman spectroscopies to touch this problem? In fact in papers by Kremer and Paluch groups it was shown that application of these techniques allows to address this issue very well, please see following papers Kipnusu et al. Soft Matter,9, 4681-4686 (2013), Kipnusu et al. , Zeitschrift für Physikalische Chemie 226, (2018), Tarnacka et al. J. Phys Chem. C, 122, 28033-28044, (2018), Minecka et al. Chem. Chem. Phys., 20, 30200-30208 (2018). I would extend introduction section to put the context of variation in molecular conformation in low and high molecular weight systems confined in pores in wider perspective as quite important problem. That for sure would strengthen the outcome of the manuscript.

Our answer: We thank the referee for this relevant comment. We have followed the suggestion of the referee to extent our introduction (line 59-71).

“Generally speaking, the confinement of macromolecules inside nanoporous material such as Al2O3 or MCM was extensively investigated in the lack of a solvent [27]. Typically, it was demonstrated that increasing the constraints by decreasing the pore size enhanced the intermolecular interaction and strongly modified their properties such as the glass transition temperature [28,29]. In addition, both the interfacial energy and the wettability play a significant role on the polymer dynamic inside the nanopore[30] as well as the molecular weight of the polymer[31]. With respect to the literature, it appears clear that the polymer behavior under constraint involves complex phenomena where numerous parameters influence their structures. According to that and considering the addition of an electrolyte solution, the question about the impact of confinement on the polymer conformation should be well understood since the keystone of the ionic transport properties. This will help the experimentalist to optimize the design of polymer functionalized nanopore.”

We agree that FTIR and Raman spectroscopy are powerful techniques to characterize the conformation of the polymer, as well as the dynamic can be investigated by complex impedance spectroscopy. However, in the present study these methods cannot be used. First, we work at single nanopore level that disqualifies the usual spectrometry method due to the low number of PEG molecules on the sample, compared to the bulk material. One solution could be to work on multipore membrane. Actually, this solution is also no-suitable since the FTIR analyses consist to investigate the H-bond of OH moieties (between 3400 cm-1 and 3600 cm-1), where there is also an absorption band of the PET and water. In that case, the FTIR cannot be used. The Raman spectroscopy could be suitable. However, working on multipore membrane generate a disparity of the pore size. Finally, our investigation aims to provide explanation of the PEG conformation under condition of nanopore sensors that the reason why we have focused our investigation at single nanopore level.

  • The other quite obvious issue that may explain data reported by Authors is variation in wettability of urea, water or PEG close to the pore walls. Did Authors consider measuring contact angles to correlate their results with wettability. Herein it should be noted that Floudas group (Alexandris et al. Macromolecules 2016, 49, 7400−) and further Talik et al. (JPC C 123, 5549−5556, 2019) have shown that interfacial energy, wettability are two important factors controlling dynamics and most likely conductivity (it is directly connected to the mobility of the ions and structural/segmental dynamics).

Our answer: We thank the referee and we agree that the interfacial energy and the wetting modified the ionic conductivity. Unfortunately, we cannot provide a rigorous measurement of the contact angle of the inner nanopore surface. Indeed, as shown in figure 2c of the manuscript, the PEG density is depended on the nanopore diameter due to the different surface chemistry induced by the track-etched process. Shortly, the bombardment with the heavy swift ion created a halo of damage. The diameter of this halo is several teens of nm that depends on the type of material and its properties (crystallinity, orientation, etc.), the energy and the type of ion. This means that after, the chemical etching does not attack the bulk material below the halo size typically in our case. Thus, a measurement of the contact angle performed on the surface of PET film functionalized with PEG after chemical etching does not make sense, since it would be different from the inner surface of the nanopore.

We have, however, mentioned the role of the surface wettability and the surface energy in the introduction section dedicate to extend the problematic of the confined polymer (line (63-64))

  • Authors did not reference to quite large and significant articles devoted to studying charge transport in pores (see papers by Kremer or Singh groups for example Iacob et al. Soft Matter 2012, 8, 289− Tripathi et al. J. Mater. Chem. A 2015, 3, 23809−23820.). I think additional discussion in view of these papers would be useful for the Authors.

Our answer: We thank the referee for this comment. Indeed, the charge transport through mesoporous porous material was widely investigated. As mentioned by the referee, we did not mention these references in our manuscript because these systems were quite far. Indeed, the referee mentioned investigations performed in the lack of solvents (or just a few molecules (typically the coordination shell)) since the impedance complex spectroscopy were conducted on dry samples. In our case, we investigate system where the charges are inside a solvent (here water). We think that mention this literature will make our message more confusing since the charge transport inside a solid or in a confined solvent obey to different laws.

Less important /minor points

  • Authors in many places use SiN acronym for pores. Please defined what SiN stands for.

Our answer: The SiN is silicon nitride, it is actually the most common material used for single nanopore sensing. We have clarified this point in the revised version of the manuscript.

  • Authors should improve quality of the Figures they are not readable.

Our answer: We have carefully redrawn all the figures. We hope they are now in correct format.

  • Authors should provide more explicit explanation why did they use urea not other agents?

Our answer: As mentioned in the introduction, the aim of our investigation was to understand how the PEG is structured and influenced the ionic transport in the case of nanopore sensing experiments. In this field, one very relevant challenge is to identify the protein conformational change or aggregation at single molecule level. The urea is one the most denaturant agent uses to denaturate a protein. However, it is not always take into account as also an agent that modifies the electrolyte transport and thus the ionic current. A comment has been added in the introduction (line 100-102)

  • Under equation 1 authors did not define what a stands for. I think it should be the size of one monomer. Please correct it.

Our answer: Referee is right. The correction has been done

After addressing following issues I do recommend publication of the paper in Nanomaterials.

Round 2

Reviewer 2 Report

I think Authors responded thoroughly to my comments. Therefore I recommend publication of the manuscript in Nanomaterials in the current form.